# Influence of Blood Contamination on Push-Out Bond Strength of Three Calcium Silicate-Based Materials to Root Dentin

Cristina Rodrigues Paulo [1], Joana A. Marques [1], Diana B. Sequeira [1], Patrícia Diogo [1], Rui Paiva [2], Paulo J. Palma [1,3] and João Miguel Santos [1,3,*]

1   Institute of Endodontics, Faculty of Medicine, University of Coimbra, 3000-075 Coimbra, Portugal; uc2014216744@student.uc.pt (C.R.P.); joanaamarques@uc.pt (J.A.M.); dianasequeira@fmed.uc.pt (D.B.S.); pdn@uc.pt (P.D.); ppalma@uc.pt (P.J.P.)
2   Polytechnic of Leiria, 2411-901 Leiria, Portugal; rui.paiva@ipleiria.pt
3   Center for Innovation and Research in Oral Sciences (CIROS), Faculty of Medicine, University of Coimbra, 3000-075 Coimbra, Portugal
*   Correspondence: jsantos@fmed.uc.pt; Tel.: +351-239-249-151

**Abstract:** A proper bond between root canal filling materials and dentin surface is essential to resist dislodgement and guarantee long-term success. Blood exposure is likely to occur in various clinical situations in which calcium silicate-based materials are used; therefore, it is fundamental to render data concerning the influence of blood on bond strength. The present study aims to evaluate the effect of blood contamination on the push-out bond strength obtained with three different biomaterials to root canal dentin; Ninety extracted human mono-radicular permanent teeth were selected. The root canals were prepared with Gates Glidden burs until a diameter of 1.10 mm was achieved. Teeth were then randomly divided into six experimental groups ($n = 15$) according to the presence/absence of blood contamination and biomaterial used for root canal filling (ProRoot® MTA, Biodentine™, and TotalFill® BC Putty). After one week, each root was sectioned in three segments (coronal, middle, and apical regions). Specimens were then submitted to push-out bond strength tests. Fracture pattern evaluation was performed. The significance level was set at 5%.; Blood contamination did not affect the push-out bond strength of any of the three tested calcium silicate-based cements ($p > 0.05$). Regardless of blood contamination, TotalFill showed statistically higher push-out bond strength when compared with Biodentine ($p = 0.040$) and MTA ($p = 0.004$). Biodentine exhibited higher bond strength than MTA ($p = 0.043$). Biomaterials' comparison within each radicular segment revealed statistically superior bond strength of both Biodentine and TotalFill over MTA ($p < 0.05$) in the coronal segment. TotalFill presented higher push-out bond strength regarding the apical segment compared to Biodentine ($p = 0.003$). Fractures were mostly adhesive.; Overall results indicate TotalFill presents the highest push-out bond strength values, followed by Biodentine and, lastly, MTA. Blood contamination did not affect the dislodgement resistance. Biomaterials' comparison within each radicular segment revealed both TotalFill and Biodentine as the preferable alternatives for application in the coronal region. TotalFill might be the biomaterial of choice for placement in the apical region.

**Keywords:** Biodentine; blood contamination; calcium silicate-based cements; mineral trioxide aggregate; push-out bond strength; TotalFill BC Putty





## 1. Introduction

Contemporary endodontics offers a wide span of materials for root canal filling, which leads to an improved endodontic treatment success rate, as novel materials exhibit better biocompatibility, bioactivity, and sealing properties [1,2]. An appropriate bond strength at the material-dentin interface is crucial to ensure a proper seal of the root canal system, with the ultimate goal of preventing or minimizing microleakage [3–5].

Mineral trioxide aggregate (MTA) was the first hydraulic cement successfully introduced in endodontics due to its biocompatibility, low solubility, radiopacity, hard-tissue

inductive and conductive activity, ability to set in a wet environment, and ability to imprison bacteria as a result of its in situ precipitation [1,6–8]. MTA presents a suitable biomaterial for an extensive range of clinical applications, including vital pulp therapy [9], regenerative endodontic procedures [10], apexification, perforation and resorption repair, as well as root-end and root canal filling [11–13]. However, this calcium silicate-based cement presents some well-known drawbacks, including its long setting time, challenging handling properties, potential for tooth discoloration, and high cost [12,14].

Aiming to overcome MTA shortcomings, several newly developed hydraulic materials were launched. Biodentine™ (Septodont, Saint-Maur-des-Fossés Cedex, France) shows mechanical properties and clinical applications comparable with MTA [15]. Conversely, this bioactive dentin substitute material presents a shorter setting time (approximately 12 min) due to the inclusion of calcium chloride as a setting accelerator within its composition alongside improved handling properties [16,17]. Moreover, Biodentine™ is associated with better esthetic outcomes, as it demonstrates superior color stability compared with MTA [14].

More recently, Endosequence Root Repair Material (ERRM; Brasseler USA, Brasseler USA, Savannah, GA) became commercially available and is now present in Europe as TotalFill® (FKG, La Chaux-de-Fonds, Switzerland), a pre-mixed and ready-to-use material provided in both scoopable or injectable versions. This biomaterial presents reasonably good handling properties and a fast initial setting time of 2 h [18]. Besides being hydrophilic, radiopaque, and having a high pH level, excellent physical and biological properties have been reported in previous studies, with data confirming its biocompatibility, cytotoxicity levels, and antibacterial properties similar to MTA [19–21].

Bioactive materials can be subjected to functional load or forces resulting from restorative material placement. Therefore, a proper bond between the material and the dentin surface is essential to resist dislodgement and guarantee long-term success [22–24]. However, blood contamination has been identified as one potential factor affecting MTA's physical properties and retention to root dentin, consequently jeopardizing its sealing ability [25–27]. In contrast, some studies found that blood did not negatively impact bond strength [4,28,29]. Since blood exposure is likely to occur in various clinical situations in which calcium silicate-based materials are used, it becomes fundamental to render data concerning the influence of blood on the bond strength obtained with more recently introduced materials, namely Biodentine™ and TotalFill® BC Putty. Furthermore, considering the discrepancy of results found in literature regarding the impact of blood contamination on the adhesion of different biomaterials to root dentin, it becomes crucial to provide additional data on the topic. Push-out bond strength tests allow assessment of the bond strength inherent to the biomaterial-root canal dentin adhesive interface [30].

The present study aims to evaluate the effect of blood contamination on the push-out bond strength obtained with three different biomaterials (ProRoot® MTA, Biodentine™, and TotalFill® BC Putty) to root canal dentin.

The null hypothesis states they have are similar push-out bond strength values regardless of blood exposure.

## 2. Materials and Methods

### 2.1. Specimen Selection

The present study was approved by the Ethics Committee of the Faculty of Medicine of the University of Coimbra (notification CE-001/2013) and followed the guidelines of the Declaration of Helsinki. A total of ninety (90) extracted human permanent teeth were selected. Sample size was defined based on previous studies with similar methodology [25,26,28,29,31]. The inclusion criteria consisted of mono-radicular teeth with mature apices, clinically and radiographically free of caries or cracks, presenting straight roots, with no previous root canal treatment. All external surfaces were cleaned using periodontal scalers to remove any soft tissue remnants and calculus. The specimens were then kept in 0.5% chloramine-T solution for a maximum of 6 months.

### 2.2. Root Canal Biomechanical Preparation

Afterward, teeth were sectioned with a low-speed diamond disc (Accutom-50; Struers, Ballerup, Denmark) under continuous water cooling to obtain a standardized 15-mm root length and an apical diameter of 2 mm. The root canals were manually instrumented up to #30 K-file (Dentsply Maillefer, Ballaigues, Switzerland), and intermittent irrigation with 1 mL of 3% sodium hypochlorite (CanalPro$^{TM}$ NaOCl 3%; Coltène/Whaledent, Altstätten, Switzerland) between instruments was ensured, totaling a volume of 3 mL of irrigant solution. Apical patency was maintained throughout instrumentation with a #20 K-file. Gates Glidden burs (Dentsply Maillefer, Ballaigues, Switzerland) sizes 1 to 4 were then used to mechanically prepare the root canal systems until a standardized diameter of 1.10 mm was achieved. Irrigation during Gates Glidden series preparation with a total of 3 mL of 3% sodium hypochlorite was also assured. A final rinse with 2 mL of 17% EDTA (CanalPro$^{TM}$ EDTA; Coltène/Whaledent, Altstätten, Switzerland) for 1 min was performed, followed by 1-min irrigation with 2 mL of saline solution (NaCl 0.9%; B. Braun Medical, Melsungen, Germany). All irrigation procedures were accomplished using disposable syringes and notched needles (Monoject, 27G; Covidien, Dublin, Ireland). The root canal space was subsequently dried using ISO 60 sterile absorbent paper points (Zipperer Absorbent Paper Points; VDW, Munich, Germany).

### 2.3. Blood Collection

Following informed consent, a blood sample (6 mL) was obtained from one healthy volunteer researcher by venipuncture. Blood collection was performed immediately before obturation procedures. The blood was kept in a sterile blood tube, internally coated with spray-dried tripotassium ethylenediaminetetraacetic acid (K2EDTA; Ref 454,086 Lot A18113BQ) to prevent clotting.

### 2.4. Root Canal Filling

Afterwards, teeth were randomly divided in six experimental groups (stratified random sampling method) according to the obturation protocol, namely regarding the presence/absence of blood and/or the biomaterial used for root canal filling, as seen in Table 1.

**Table 1.** Study groups.

| Study Group<br>*n* = 15 | Biomaterial | Blood Contamination |
|---|---|---|
| MTA/Blood | ProRoot MTA | Yes |
| MTA/Saline | ProRoot MTA | No |
| Biodentine/Blood | Biodentine | Yes |
| Biodentine/Saline | Biodentine | No |
| TotalFill/Blood | TotalFill BC Putty | Yes |
| TotalFill/Saline | TotalFill BC Putty | No |

All bioactive materials were prepared according to the manufacturer's instructions (Table S1). In MTA/Blood, Biodentine/Blood, and TotalFill/Blood groups, 7 µL of the freshly collected blood were firstly dropped in the root canal, and immediately after, biomaterials were applied. In MTA/Saline, Biodentine/Saline, and TotalFill/Saline groups, no blood contamination was performed. The specimens were filled with either ProRoot MTA (MTA/Blood and MTA/Saline), Biodentine (Biodentine/Blood and Biodentine/Saline), or TotalFill BC Putty (TotalFill/Blood and TotalFill/Saline). The former was prepared on a sterilized glass plaque until achieving a sandy consistency. Biodentine is available in pre-dosed capsules for mixing, whereas TotalFill BC Putty presents as a ready-to-use scoopable material. All biomaterials were delivered into the root canal using the MAP$^{TM}$ System A0662-0 conveyor (Dentsply Maillefer, Ballaigues, Switzerland), and vertical pluggers (1/2 and 3/4 Machtou pluggers; Denstply Maillefer, Ballaigues, Switzerland) were subsequently used according to the diameter of the root canal region for bioactive materials compaction.

Root canal filling procedure was performed under 10x magnification (Leica Microscope M320; Leica Microsystems, Heerbrugg, Switzerland). Following obturation, teeth were stored in an incubator at 37 °C and 100% relative humidity for one week to allow complete setting of the materials.

Hereafter, samples were sectioned perpendicularly to the longitudinal axis of the root using a water-cooled precision diamond saw M1D10 under low intensity at 1000 rpm and 0.05 mm/second speed (Accutom-50; Struers, Ballerup, Denmark). Three sections corresponding to coronal, middle, and apical regions with $3 \pm 0.1$ mm thickness were obtained from each root, providing a total of 270 samples. The measurement of each section was carried out using a digital caliper (156-105-10; Mitutoyo, Kawasaki, Japan) with 0.01 mm precision to guarantee a proper thickness.

All previously described experimental procedures (Figure 1) were performed at 21.8 °C room temperature and 44% humidity by a single experienced operator.

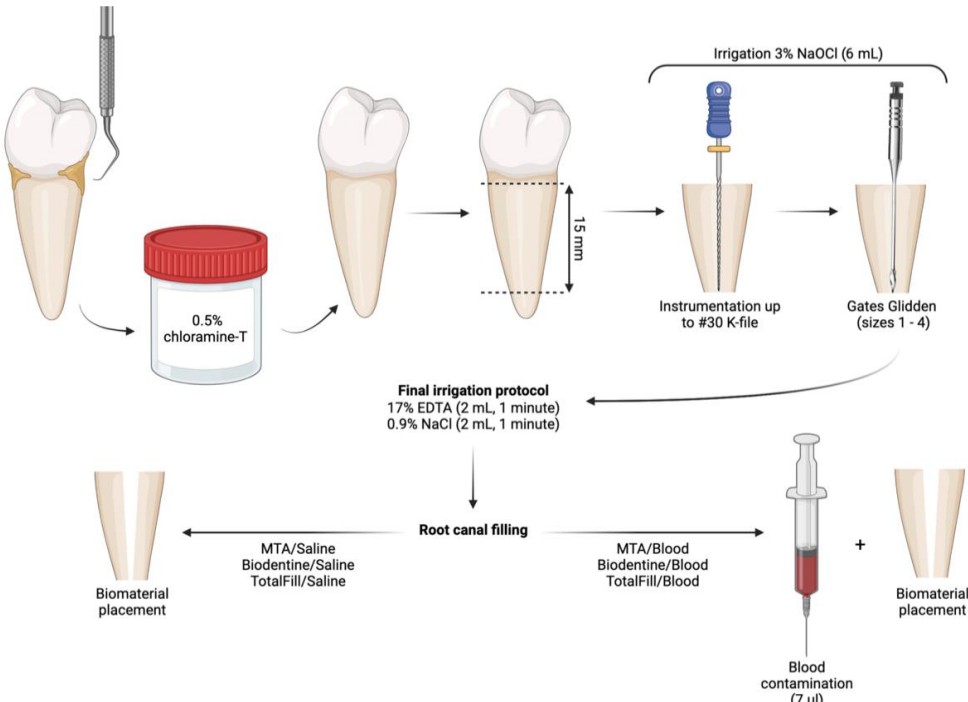

**Figure 1.** Experimental protocol schematic representation. Created with BioRender.com.

## 2.5. Push-Out Bond Strength Test

The sequence of testing was randomly defined. The assessment of the push-out bond strength was carried out using a universal testing machine (model AG-I, Shimadzu Corporation, Kyoto, Japan). Each specimen was set up on an aluminum cylinder with a central hole, and a flowable composite resin (SDR; Dentsply Detrey GmgH, Konstanz, Germany) was used to hold the positioning during testing (Figure 2). The compressive load was applied by exerting an apical-coronal directed pressure on the surface of the endodontic material using a 0.9 mm diameter plugger at a crosshead speed of 0.5 mm/min. The bur had a clearance from the margin of the dentinal wall to ensure contact with endodontic material only. The maximum load (N) applied at the time of dislodgement was recorded. Afterward, the push-out bond strength values (MPa) were calculated by dividing the maximum load by the total adhesion area (mm$^2$).

Following photographic record with a Canon EOS 5DsR camera (Canon EF 100 mm f/2.8L Macro IS USM Lens), the radius values of the root canals from both coronal and apical sides of each section were assessed using ImageJ Software (ImageJ v1.52, National Institutes of Health, Bethesda, MD, USA), in order to determine the adhesive interface area. Since the root canals present an internal taper similar to a truncated cone, the formula for

calculating the cone's lateral surface area was used (Figure S1A). Thus, the area of interest was provided by subtracting the lateral area of a smaller cone from the total area of a cone with larger dimensions. Since only the radius and height of the truncated cone are obtained through measuring, the total height is attained by resorting to the principle of triangles similarity, and the slant height is derived from the Pythagorean theorem (Figure S1B–D).

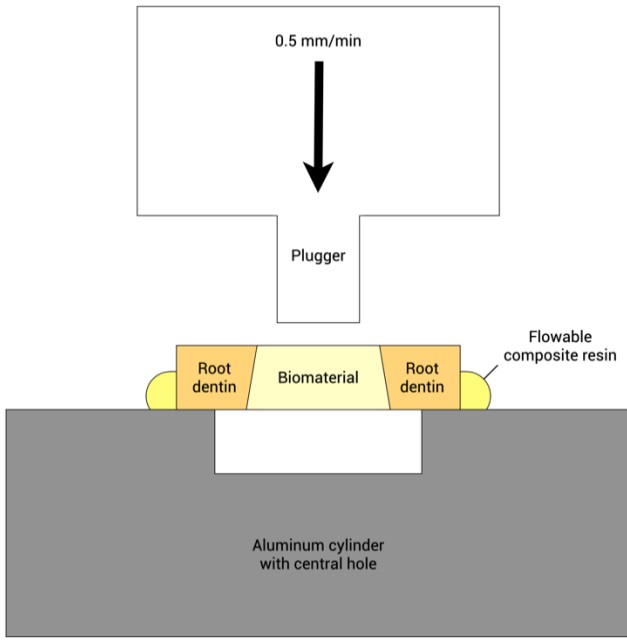

**Figure 2.** Schematic representation of the push-out bond strength test set-up (*arrow*: compressive load direction).

### 2.6. Fracture Pattern Evaluation

Samples were examined under 8× magnification using a stereomicroscope (Nikon SMZ1500, Tokyo, Japan) to categorize the fracture pattern as follows: (1) adhesive (dentin-biomaterial interface), (2) cohesive (within the biomaterial), and mixed (combination of both adhesive and cohesive patterns).

### 2.7. Statistical Analysis

Statistical analysis was performed using R Commander (version 2.7-1 Boca Raton, FL, USA). Normality of data distribution was assessed with Shapiro–Wilk test. Subsequently, Kruskal–Wallis testing was carried out to assess differences between all groups regarding dislodgement force. As long as $n \geq 30$, *t*-test for independent samples allowed biomaterials comparison. Afterward, F test for equality of variances was used to compare both tested conditions (presence/absence of blood) for each material. Biomaterials' comparison within each radicular segment was carried out using the following statistical tests after normality assessment with Shapiro–Wilk test: Kruskal–Wallis followed by Wilcoxon testing (coronal region), ANOVA (middle region), or ANOVA followed by *t*-test for independent samples (apical region). The significance level was set at 5%.

## 3. Results

### 3.1. Push-Out Bond Strength

Blood contamination did not affect the push-out bond strength of any of the three tested calcium silicate-based cements ($p > 0.05$). The distribution of push-out bond strength values within each experimental group is shown in Figure 3. TotalFill presented the highest mean push-out bond strength value regardless of blood contamination, with statistically significant differences being detected when compared with both Biodentine ($p = 0.040$) and

MTA ($p = 0.004$). Additionally, statistical differences were found between Biodentine and MTA ($p = 0.043$), with the former exhibiting higher bond strength values (Table 2).

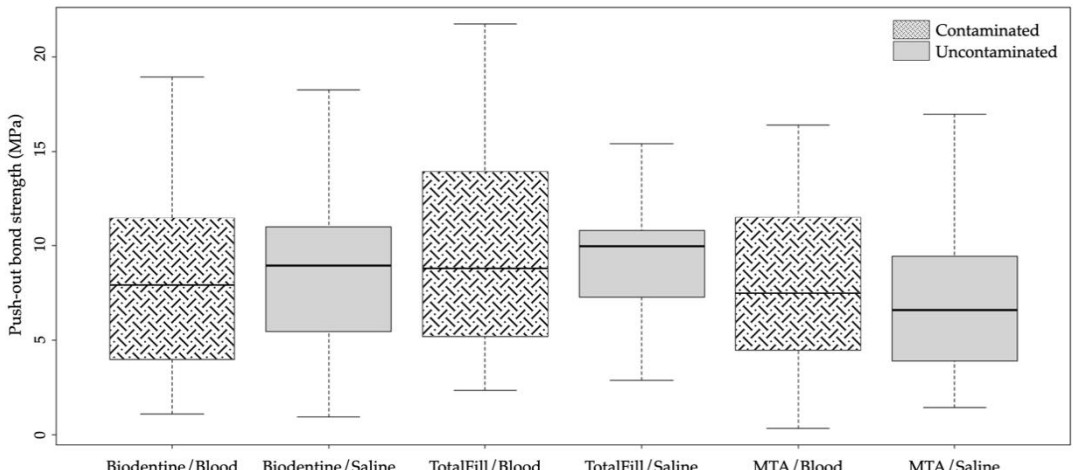

**Figure 3.** Box-plot of push-out bond strength distribution within each experimental group, according to both biomaterial and condition (presence or absence of blood contamination).

**Table 2.** Push-out bond strength descriptive statistics for each biomaterial regardless of blood contamination (MPa).

| Biomaterial | Mean $\pm$ SD | Median |
|---|---|---|
| ProRoot MTA | 7.44 $\pm$ 4.07 | 6.83 |
| Biodentine | 8.54 $\pm$ 4.49 | 8.62 |
| TotalFill BC Putty | 10.32 $\pm$ 8.41 | 9.97 |

SD, standard deviation.

Biomaterials' comparison within each radicular segment (Table 3) revealed statistically significant differences between MTA and both Biodentine ($p < 0.001$) and TotalFill ($p = 0.008$) in the coronal segment. No statistical differences were found in the middle region between the different biomaterials ($p > 0.05$). Regarding the apical segment, TotalFill presented statistically higher mean push-out bond strength when compared to Biodentine ($p = 0.003$), with no statistical differences being detected between MTA and the remaining two biomaterials ($p > 0.05$).

**Table 3.** Push-out bond strength values (MPa) according to radicular regions.

| Biomaterial | Radicular Segment | | |
|---|---|---|---|
| | Coronal (Median) | Middle (Mean $\pm$ SD) | Apical (Mean $\pm$ SD) |
| ProRoot MTA | 4.97 [a] | 8.22 $\pm$ 3.34 [a] | 7.98 $\pm$ 4.28 [a,b] |
| Biodentine | 9.10 [b] | 9.54 $\pm$ 5.53 [a] | 6.83 $\pm$ 3.68 [a] |
| TotalFill BC Putty | 8.64 [b] | 10.77 $\pm$ 4.95 [a] | 9.56 $\pm$ 3.81 [b] |

Groups sharing the same superscript letter within each radicular segment do not present statistically significant differences ($p > 0.05$) according to Kruskal–Wallis followed by Wilcoxon testing (coronal region), ANOVA (middle region), or ANOVA followed by *t*-test for independent samples (apical region).

### 3.2. Fracture Pattern

Fractures were mostly adhesive regardless of the biomaterial and blood contamination (Table 4). One cohesive fracture was recorded in the Biodentine/Saline group. Moreover, mixed failures were found in TotalFill, MTA, and Biodentine specimens (11.1%, 10% and 6.7%, respectively).

**Table 4.** Fracture pattern analysis results.

| | Fracture Pattern | | |
|---|---|---|---|
| **Experimental Group** | **Adhesive** | **Cohesive** | **Mixed** |
| MTA/Blood | 39 | 0 | 6 |
| MTA/Saline | 42 | 0 | 3 |
| Biodentine/Blood | 45 | 0 | 8 |
| Biodentine/Saline | 38 | 1 | 6 |
| TotalFill/Blood | 43 | 0 | 2 |
| TotalFill/Saline | 37 | 0 | 8 |

## 4. Discussion

This ex vivo study intended to assess the influence of blood contamination on the push-out bond strength obtained with three different bioactive materials to root canal dentin. The ability to ensure a proper seal is one of the major requirements for root canal filling materials [5]. In many clinical applications, including vital pulp therapy, regenerative endodontic procedures, perforation and resorption repair, apical microsurgery, and apexification procedures, calcium silicate-based cements often undergo blood contamination even before completion of their setting reactions. Regardless of the context, adequate bond strength at the biomaterial-root canal dentin interface is crucial to withstand dislodgement forces during the immediate or delayed restorative procedures and resist masticatory forces.

Among several currently available techniques, push-out bond strength testing is widely recognized as the most reliable method to assess the dislodgement resistance of materials to root dentin [28,32]. Chen et al. demonstrated that to apply the push-out test correctly, sample thickness should be greater than 1.10 mm, and the ratio between the plugger and filling material diameter should be no greater than 0.85. In our study, both recommendations were followed [30]. Furthermore, in the present study, bond strength testing was carried out considering the worst-case scenario since the load during testing was exerted in apical-coronal direction, therefore simulating an unfavorable root canal geometry. Thus, the risk that eventual mechanical retentions could lead to an overestimation of bond strength was minimized.

The effectiveness of bonding to root canal dentin relies, among several factors, on the selected biomaterial. The initial seal obtained with bioactive materials is purely mechanical. However, subsequent biomineralization occurs at the biomaterial-dentin interface through hydroxyapatite crystals formation over time, providing it with chemical bonding [23,33–35]. In the present experimental study, TotalFill presented statistically higher mean push-out bond strength values when compared with both Biodentine and MTA, which agrees with Kadic et al. [36]. A possible explanation relies on TotalFill's nanospheric size particles (maximum 1e-3 μm) that promote, for the same volume of material, an increased surface area in contact with tissue fluids. Consequently, higher carbonated apatite precipitation is expected, with subsequent superior formation of tag-like structures [35]. The more favorable consistency and handling properties of TotalFill over the two remaining tested materials might also explain its superior results and contribute to greater outcome predictability.

Moreover, statistically higher push-out bond strength was verified for Biodentine over MTA, which is in agreement with both Akcay et al. and Marquezan et al. in their previous studies [25,31]. The addition of calcium chloride to Biodentine's composition seems to act as a catalyst for tricalcium silicate hydration [17]. During hydration, small fibrous crystal formation occurs, which indicates rapid crystallization. This phenomenon might arise from the high ionic content of the solution when calcium chloride is present during tricalcium silicate setting and hardening, leading to the formation of heterogeneous density areas instead of a homogeneously dense structure. This facilitates the diffusion of ions and water through the initial calcium silicate layer, thus allowing a higher rate of hydration during the early diffusion-controlled period [37]. Additionally, the superior

content of calcium-releasing products, verified with Biodentine when compared to MTA, may enhance biomineralization and, consequently, magnitude of the bond strength [25,35].

In addition, it is important to underline the considerably higher standard deviation verified in TotalFill specimens ($10.32 \pm 8.41$) when compared with both Biodentine ($8.54 \pm 4.49$) and MTA ($7.44 \pm 4.07$). Therefore, although TotalFill presented statistically higher mean bond strength value, a higher standard deviation might reflect a lower outcome predictability in regard to retention of the biomaterial. In the present study, all experimental procedures were performed by a single experienced operator under identical environmental conditions and testing time as well as similar adhesion area, with the ultimate goal of minimizing the impact of these variables on the results. We also hypothesize these findings to be related with dentin being an extremely heterogeneous substrate, although teeth exhibiting obvious root dentin alterations have been discarded [38]. In addition, based on the obtained results, a superior sample size should be considered in future studies.

As previously described, the push-out bond strength of three hydraulic cements was assessed in two distinct conditions regarding the setting environment: presence and absence of blood. Our findings show that blood contamination did not affect the retention of the tested biomaterials ($p > 0.05$), with uncontaminated and contaminated specimens of each biomaterial exhibiting similar bond strength values. Therefore, the null hypothesis has been accepted. Accordingly, Adl et al. found that blood contamination did not adversely affect MTA and CEM bond strength [29]. Moreover, Marquezan et al. found no differences between uncontaminated versus contaminated Biodentine specimens regardless of push-out bond strength assessment time (24 h, 7 days, or 28 days), whereas MTA-Angelus bond strength was negatively affected over time [31]. Although blood exposure has also been identified as a factor that impairs biomaterials' dislodgement resistance [25,26], we speculate our results to be related to the bioactive materials' hydrophilicity. Even though specimens were kept hydrated with saline solution throughout the entire experimental period, blood's presence in the initial setting of the biomaterials may have mitigated a possible negative impact [29]. Moreover, the affinity of red blood cells to type I collagen (a major component of dentin's organic phase) is expected to promote dentinal tubules occlusion, creation of gaps, blocking tag-like structures formation, and, consequently, impairing bond strength [4,28,39]. However, the penetration depth of biomaterials within the dentinal tubules is solely responsible for the micromechanical anchorage of the cement. By offering a moist setting environment, blood may have contributed to higher bioactivity, with greater apatite precipitation and, consequently, bonding [29]. Despite the results of this study and considering the heterogeneity of scientific evidence regarding the role played by blood on biomaterials' adhesion, ensuring complete hemostasis should be considered an essential prelude towards a favorable outcome of the endodontic treatment [9,14,21,27–29].

Biomaterials' comparison according to root segment variation was also carried out. MTA yielded the lowest push-out bond strength values in the coronal region, with statistical differences observed compared with TotalFill and Biodentine. Conversely, similar bond strength results were found between TotalFill and Biodentine, meaning that in a clinical scenario in which regenerative endodontic procedures requiring the creation of a cervical barrier are performed, the use of either TotalFill or Biodentine should be recommended over MTA. Furthermore, although no statistical differences were found between the tested materials in the middle region, TotalFill exhibited statistically higher bond strength when compared with Biodentine in the apical segment, with no differences being detected between both aforementioned materials and MTA. Possible clinical implications rendered by these findings include the suggestion of TotalFill as a potential biomaterial of choice for apexification procedures in which the placement of an apical barrier is involved. Additionally, the resemblance of bond strength among all tested hydraulic cements in the middle region may uphold that all materials represent suitable alternatives for application in the referred root third.

Fracture pattern analysis mostly showed adhesive fractures in all experimental groups, regardless of the setting environment, occurring at the material-dentin interface. These findings indicate that the bond strength between the calcium-silicate based cements and root dentin surface is lower than the cohesive strength of the biomaterials. The clear predominance of adhesive failures might reflect biomaterials adequately performed biomaterials mixing, with correct underlying proportions, and successful three-dimensional root canal filling. Moreover, as abovementioned, hydration significantly improves push-out bond strength, and under moist conditions, previous reports indicate bond strength to increase from day 3 to 21 [29,40]. Thus, we hypothesize the adhesive mode of failure to possibly be related with the short storage time that preceded bond strength testing, which we found to be equal or even smaller in several published studies [4,25,26,28]. Since it is suggested that the formation of chemical bonding leads to dentin-hydraulic cements bonding enhancement over time [23], further long-term studies are needed to evaluate the effect of aging on biomaterial-dentin adhesion.

### 5. Conclusions

Within the limitations of this ex vivo study, the following conclusions can be drawn:

- Overall results indicate TotalFill presents the highest push-out bond strength values, followed by Biodentine and, lastly, MTA.
- Blood contamination did not affect the dislodgement resistance regardless of the tested hydraulic cement.
- Biomaterials' comparison within each radicular segment revealed both TotalFill and Biodentine as the preferable alternatives for application in the coronal region. Although all biomaterials present similar bond strength values in the middle third, TotalFill might be the biomaterial of choice for placement in the apical region.

**Supplementary Materials:** The following are available online at https://www.mdpi.com/article/10.3390/app11156849/s1, Figure S1: (A) Formula for cone's lateral surface area calculation; (B) Triangles' similarity (1—Representation of root canal taper; 2—Schematic representation of triangle similarity; 3—Specimen representative longitudinal section); (C) Pythagorean theorem; (D) Adhesion area formula. Table S1: Calcium silicate-based materials specifics.

**Author Contributions:** Conceptualization, P.J.P. and J.M.S.; methodology, C.R.P., D.B.S. and J.M.S.; software, C.R.P. and R.P.; validation, J.A.M., P.D. and J.M.S.; formal analysis, J.A.M., R.P. and J.M.S.; investigation, C.R.P., P.J.P. and J.M.S.; resources, J.M.S.; data curation, C.R.P. and J.A.M.; writing—original draft preparation, C.R.P., J.A.M., R.P. and J.M.S.; writing—review and editing, J.A.M., D.B.S., P.D., P.J.P. and J.M.S.; visualization, J.A.M., R.P. and J.M.S.; supervision, P.J.P. and J.M.S.; project administration, J.M.S.; funding acquisition, J.M.S. All authors have read and agreed to the published version of the manuscript.

**Funding:** This research received no external funding.

**Institutional Review Board Statement:** The study was conducted according to the guidelines of the Declaration of Helsinki, and approved by the Ethics Committee) of the Faculty of Medicine of the University of Coimbra (notification CE-001/2013).

**Informed Consent Statement:** Informed consent was obtained from all subjects involved in the study.

**Data Availability Statement:** Data available on request by contacting the corresponding author.

**Conflicts of Interest:** The authors declare no conflict of interest.

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
