# Peer review of "Influence of Blood Contamination on Push-Out Bond Strength of Three Calcium Silicate-Based Materials to Root Dentin"

_applsci, doi:10.3390/app11156849_

Round 1

Reviewer 1 Report

There are some weaknesses through the manuscript which need improvement. Therefore, the submitted manuscript cannot be accepted for publication in this form, but it has a chance of acceptance after a major revision. My comments and suggestions are as follows:

1- Abstract gives information on the main feature of the performed study, but a couple of sentences about background of this study must be added.

2- Authors must clarify necessity of the performed research. Objectives of the study must be clearly mentioned in introduction.

3- The literature study must be enriched. In this respect, authors must read and refer to the following papers: (a) https://doi.org/10.1016/j.jmbbm.2019.02.009 (b) https://doi.org/10.1016/j.jmbbm.2018.02.029 The current version of introduction is too short.

4- It would be nice, if authors could add some schematic figures to show concept and some conditions.

5- How the values presented in Table 2 and 3 were obtained? Details of calculations must be presented.

6- Although the manuscript deals with experimental practice, there is no figure to show and prove test conditions. It is necessary to add figure in several sections and subsection. For instance, authors must add figure to 3.2 to show different failure modes which are summarized in Table 4.

7- Reason for the deviation presented in the results must be discussed. In addition, error in calculation must be considered and discussed.

8- In its language layer, the manuscript should be considered for English language editing. There are sentences which have to be rewritten.

9- The conclusion must be more than just a summary of the manuscript. List of references must be updated based on the proposed papers.

10- In section 2 it has been mentioned "The specimens were then kept in 91 0.5% chloramine-T solution for a maximum of 6 months". How this condition was selected?

11- It is necessary to add figures in subsection 2.6 to show fracture pattern in examined specimens.

12- Details of experiments on determining bond strength must be added.

Please provide all changes by red color in the revised version.

Author Response

Cover letter

Point-by-Point Response to Reviewers

The authors confirm Figure 1 is original (fully composed by the authors using BioRender.com).

Reviewer #1:

There are some weaknesses through the manuscript which need improvement. Therefore, the submitted manuscript cannot be accepted for publication in this form, but it has a chance of acceptance after a major revision. My comments and suggestions are as follows:

  1. Abstract gives information on the main feature of the performed study, but a couple of sentences about background of this study must be added.

Author’s response: We thank reviewer #1 for the valuable suggestion. Accordingly, improvement of the background of the study within the Abstract section was performed.

Revised text: “A proper bond between root canal filling materials and dentin surface is essential to resist dislodgement and guarantee long-term success. Blood exposure is likely to occur in various clinical situations in which calcium silicate-based materials are used, therefore becoming fundamental to render data concerning the influence of blood on bond strength. The present study aims to evaluate the effect of blood contamination on the push-out bond strength obtained with three different biomaterials to root canal dentin;” – page 1, lines 13-18.

  1. Authors must clarify necessity of the performed research. Objectives of the study must be clearly mentioned in introduction.

Author’s response: Once again, we thank reviewer #1 for the comments. In fact, the relevance of the study, as well as the objective of the research, are presented in the last paragraphs of the Introduction (page 2, lines 80-93 and 94-96, respectively), immediately followed by the null hypothesis statement. However, following the reviewer’s comment, the relevance of the study has been improved.

Revised text: “However, blood contamination has been identified as one potential factor affecting MTA’s physical properties and retention to root dentin, consequently jeopardizing its sealing ability [25-27]. In contrast, some studies found that blood did not negatively impact bond strength [4,28,29]. Since blood exposure is likely to occur in various clinical situations in which calcium silicate-based materials are used, it becomes fundamental to render data concerning the influence of blood on the bond strength obtained with more recently introduced materials, namely BiodentineTM and TotalFill® BC Putty. Furthermore, considering the discrepancy of results found in literature regarding the impact of blood contamination on the adhesion of different biomaterials to root dentin, it becomes crucial to provide additional data on the topic. Push-out bond strength tests allow to assess the bond strength inherent to the biomaterial-root canal dentin adhesive interface [30].” – page 2, lines 83-93.

  1. The literature study must be enriched. In this respect, authors must read and refer to the following papers: (a) https://doi.org/10.1016/j.jmbbm.2019.02.009 (b) https://doi.org/10.1016/j.jmbbm.2018.02.029. The current version of introduction is too short.

Author’s response: The following article was cited in the Discussion section: https://doi.org/10.1016/j.jmbbm.2018.02.029. Concerning the inclusion of the other paper, we believe there is no relevance in adding such information since the researched topic seems not to be related with the scope of our study. The authors consider that the Introduction includes the relevant points to frame the study in the existing literature and substantiate its clinical relevance. However, the controversy of results found in the literature regarding the impact of blood contamination on bond strength, which was already mentioned in the Discussion section, was now also addressed in the Introduction.

Revised text: “However, blood contamination has been identified as one potential factor affecting MTA’s physical properties and retention to root dentin, consequently jeopardizing its sealing ability [25-27]. In contrast, some studies found that blood did not negatively impact bond strength [4,28,29]. Since blood exposure is likely to occur in various clinical situations in which calcium silicate-based materials are used, it becomes fundamental to render data concerning the influence of blood on the bond strength obtained with more recently introduced materials, namely BiodentineTM and TotalFill® BC Putty. Furthermore, considering the discrepancy of results found in literature regarding the impact of blood contamination on the adhesion of different biomaterials to root dentin, it becomes crucial to provide additional data on the topic. Push-out bond strength tests allow to assess the bond strength inherent to the biomaterial-root canal dentin adhesive interface [30].” – page 2, lines 83-93.

  1. It would be nice, if authors could add some schematic figures to show concept and some conditions.

Author’s response: We thank the reviewer #1 feedback which objective was to enhance the quality of the manuscript. Following the reviewer’s recommendation, an additional schematic representation was included to visually clarify the conditions in which push-out bond strength tests were performed.

Revised text: Figure 2 was added (page 5). Consequently, previous Figure 2 is now identified as Figure 3 (page 6).

  1. How the values presented in Table 2 and 3 were obtained? Details of calculations must be presented.

Author’s response: Table 2 presents the descriptive statistics (namely mean, standard deviation and median) regarding the push-out bond strength obtained for each biomaterial. Although it is mentioned in the main text, since the reviewer’s question suggests that we have not been clear enough in our manuscript, we also specified in the table legend that the values included in Table 2 were calculated independently of blood contamination (meaning that all samples from each material, with and without blood contamination, were considered for all calculations). Also, according to the reviewer’s instructions, we have proceeded with the incorporation of the statistical tests in Table 3 footnote.

Revised text:

Table 2. Push-out bond strength descriptive statistics for each biomaterial, regardless of blood contamination (MPa).” – page 6.

“Groups sharing the same superscript letter within each radicular segment do not present statistically significant differences (p > 0.05) according to Kruskal Wallis followed by Wilcoxon testing (coronal region), ANOVA (middle region) or ANOVA followed by t-test for independent samples (apical region).” – Table 3 footnote, page 7.

  1. Although the manuscript deals with experimental practice, there is no figure to show and prove test conditions. It is necessary to add figure in several sections and subsection. For instance, authors must add figure to 3.2 to show different failure modes which are summarized in Table 4.

Author’s response: Subsections 2.1. to 2.4. are illustrated in Figure 1. Based on the reviewer’s comments, a schematic representation (Figure 2) was included to visually clarify the conditions in which push-out bond strength tests were performed (corresponding to subsection 2.5.). Regarding fracture patterns, we present below an example of the most common failure mode (adhesive type). However, we think the inclusion of the image does not add relevant information to the paper.

Revised text: Figures 2 was included.

  1. Reason for the deviation presented in the results must be discussed. In addition, error in calculation must be considered and discussed.

Author’s response: Discussion of standard deviation was added as requested by reviewer #1.

Revised text: “In addition, it is important to underline the considerably higher standard deviation verified in TotalFill specimens (10.32 ± 8.41) when compared with both Biodentine (8.54 ± 4.49) and MTA (7.44 ± 4.07). Therefore, although TotalFill presented statistically higher mean bond strength value, a higher standard deviation might reflect a lower outcome predictability in regard to retention of the biomaterial. In the present study, all experimental procedures were performed by a single experienced operator, under identical environmental conditions and testing time, as well as similar adhesion area, with the ultimate goal of minimizing the impact of these variables on the results. We also hypothesize these findings to be related with dentin being an extremely heterogeneous substrate, although teeth exhibiting obvious root dentin alterations have been discarded [38]. Also, based on the obtained results, a superior sample size should be considered in future studies.” – page 8, lines 388-399.

  1. In its language layer, the manuscript should be considered for English language editing. There are sentences which have to be rewritten.

Author’s response: A very careful and attentive revision was performed throughout the manuscript regarding bothlanguage and form by an English language native speaker.

Revised text: Modifications were made in order to improve English language (highlighted in the manuscript).

  1. The conclusion must be more than just a summary of the manuscript.

Author’s response: We thank reviewer #1 for the comment. However, we believe the conclusions do not only summarize the obtained results, but also address the possible clinical applications of the study’s results, namely by pointing the preferable biomaterials to apply depending on the clinical scenario and by referring that blood contamination does not adversely affect the outcome of the treatment in terms of the obtained bond strength.

Revised text: Not applicable.

  1. List of references must be updated based on the proposed papers.

Author’s response: Following the reviewer instructions, one of the proposed papers was included.

Revised text: Paper https://doi.org/10.1016/j.jmbbm.2018.02.029 was added to the list of references (38).

  1. In section 2 it has been mentioned "The specimens were then kept in 0.5% chloramine-T solution for a maximum of 6 months". How this condition was selected?

Author’s response: We thank reviewer #1 for this question. The referred storage conditions were defined based on ISO 11405:2015 (Dental materials – testing of adhesion to tooth structure), according to which “Teeth that have been extracted for longer than six months may undergo degenerative changes in dentinal protein” and therefore should not be used for bond strength testing. Six months is, thus, the maximum storage time specified in the guidelines.

Revised text: Not applicable.

  1. It is necessary to add figures in subsection 2.6 to show fracture pattern in examined specimens.

Author’s response: The answer to this question was presented in previous question 6.

Revised text: Not applicable.

  1. Details of experiments on determining bond strength must be added.

Author’s response: Determination of push-out bond strength is detailed in subsection 2.5., initiating with the description of the push-out bond strength tests set up and settings (page 4, lines 214-222), followed by the indication of the formula used to calculate de push-out bond strength (page 5, lines 222-228) and the considered units. As referred in the manuscript, the push-out bond strength values were calculated by dividing the maximum load by the total adhesion area, with the maximum load being automatically provided by the universal testing machine at the time of dislodgement. The steps to determine the total adhesion area of each specimen are briefly described on page 4, lines 229-239, with all used specific calculation formulas being available in Figure S1. However, in order to clarify the conditions in which push-out bond strength tests were performed, a schematic representation of the set up was added.

Revised text: Figure 2 was included.

Reviewer 2 Report

It is a article that provides good information about new MTA materials that can be applied to root canal treatment.

Please review the following.

When describing the experimental method, there is no description of the load cell under the conditions of the push-out bond strength test. In addition, I think that  0.9mm Diameter plugger was used, which is questionable considering the root canal diameter of the teeth used in the experiment.  A more detailed explanation is required.(from line number 152)

Author Response

Reviewer #2:

It is a article that provides good information about new MTA materials that can be applied to root canal treatment. Please review the following.

  1. When describing the experimental method, there is no description of the load cell under the conditions of the push-out bond strength test. In addition, I think that 0.9mm diameter plugger was used, which is questionable considering the root canal diameter of the teeth used in the experiment. A more detailed explanation is required (from line number 152).

Author’s response: We thank reviewer #2 for the relevant risen points. The reviewer’s comment regarding the conditions in which the push-out bond strength tests were performed suggests that we have not been clear enough in our manuscript. Therefore, an additional schematic representation was included to visually clarify the set up of push-out bond strength testing. Concerning the plugger diameter of 0.9 mm, as discussed in the Discussion section, the ratio between the plugger diameter and the diameter of the biomaterial being tested should be no greater than 0.85. Since all root canals were prepared until a standardized diameter of 1.10 mm was achieved, a maximum ratio of 0.82 was verified and, thus, the current recommendations regarding appropriate selection of the plugger diameter were followed in the present study.

Revised text: Figure 2 was included.

Reviewer 3 Report

1) The author should present the stereo-photomicrographs of failure modes after push-out test.

2) The p-value must be specified in the fracture analysis table.

Author Response

Cover letter

Point-by-Point Response to Reviewers

The authors confirm Figure 1 is original (fully composed by the authors using BioRender.com).

Reviewer #3:

  1. The author should present the stereo-photomicrographs of failure modes after push-out test.

Author’s response: This question was addressed in response to reviewer 1 question 6.

  1. The p-valuemust be specified in the fracture analysis table.

Author’s response: This question was addressed in response to reviewer 1 question 5.

Reviewer 4 Report

  1. Please explain the method of Biomaterial with blood contamination in detail. Do you mean after filling the root canal with biomaterial, then the material-filled samples were soaked in blood tube?
  2. You explained the material-filled samples were sectioned perpendicularly to the longitudinal axis using a water-cooled precision diamond saw. I wonder the experimental biomaterials might be loss during this procedure.
  3. Complete setting of calcium silicate-based cement can not be achieved with blood contamination condition. If you tested the hardness of each group using Vickers hardness test, then more reliable results may be presented.
  4. In my lab test, Biodentine and BC RRM putty (TotalFill putty in your study) might be released in the growth media or osteogenic media after setting; whereas, ProRoot MTA was not released after setting. Please consider this point in comparison with your results.

Author Response

Reviewer #4:

  1. Please explain the method of Biomaterial with blood contamination in detail. Do you mean after filling the root canal with biomaterial, then the material-filled samples were soaked in blood tube?

Author’s response: This reviewer’s question suggests that we have not been clear enough in our manuscript. Therefore, manuscript modifications were performed in order to clarify this point. The freshly collected blood was firstly applied inside the root canals and immediately after the biomaterials were placed, similarly to what happens in the clinical context.

Revised text: “In MTA/Blood, Biodentine/Blood and TotalFill/Blood groups, 7 μL of the freshly collected blood were firstly dropped in the root canal and, immediately after, biomaterials were applied. In MTA/Saline, Biodentine/Saline and TotalFill/Saline groups no blood contamination was performed.” – page 3, lines 174-177.

  1. You explained the material-filled samples were sectioned perpendicularly to the longitudinal axis using a water-cooled precision diamond saw. I wonder the experimental biomaterials might be loss during this procedure.

Author’s response: We thank reviewer #3 for the comments. However, following root canal filling, all specimens were stored in an incubator at 37ºC and 100% relative humidity for one week to allow complete setting of the biomaterials. Even though the loss of the biomaterials during sectioning could be a concern, it was not verified in the present study.

Revised text: Not applicable.

  1. Complete setting of calcium silicate-based cement can not be achieved with blood contamination condition. If you tested the hardness of each group using Vickers hardness test, then more reliable results may be presented.

Author’s response: Once again, we thank reviewer #3 for the comments. In fact, the reduction of MTA’s compressive strength and a prolonged setting time after contamination with fresh human blood has been reported. The authors
were meticulous in ensuring contamination of the specimens with freshly collected blood, in order to replicate as closely as possible what happens in the clinical scenario. Although the evaluation of biomaterials’ microhardness using Vickers test would provide interesting data, this analysis was not carried out in the present study indeed. However, considering fractures were mostly adhesive in all experimental groups, the cohesive strength of all three tested calcium silicate-based cements is still superior to the bond strength between these materials and root canal dentin. This reviewer #3 suggestion will be taken into account in future perspectives of this ongoing research line.

Revised text: Not applicable.

  1. In my lab test, Biodentine and BC RRM putty (TotalFill putty in your study) might be released in the growth media or osteogenic media after setting; whereas, ProRoot MTA was not released after setting. Please consider this point in comparison with your results.

Author’s response: A recently published study carried out in a dog model confirmed the sealing ability of the three biomaterials tested in the present study in the presence of blood. Moreover, absence of both macrophagic activity and microleakage confirms an adequate clinical performance in terms of ensuring a proper seal and bioactivity (https://doi.org/10.1016/j.joen.2021.06.018). Concerning the release of ProRoot MTA and Biodentine, data from our lab shows that both materials present significant solubility (https://doi.org/10.3390/jfb9040074).

Revised text: Not applicable.

Round 2

Reviewer 1 Report

The paper has been improved and corresponding modifications have been conducted.